# Humanized Mouse Models for the Study of Infection and Pathogenesis of Human Viruses

**DOI:** 10.3390/v10110643

**Published:** 2018-11-17

**Authors:** Fritz Lai, Qingfeng Chen

**Affiliations:** 1Institute of Molecular and Cell Biology, Agency for Science, Technology and Research (A*STAR), Singapore 138673, Singapore; mdcflsc@nus.edu.sg; 2Department of Medicine, Yong Loo Lin School of Medicine, National University of Singapore, Singapore 119228, Singapore; 3Department of Physiology, Yong Loo Lin School of Medicine, National University of Singapore, Singapore 119228, Singapore

**Keywords:** humanized mice, viral infections, human immune system, liver chimeras, human specific responses

## Abstract

The evolution of infectious pathogens in humans proved to be a global health problem. Technological advancements over the last 50 years have allowed better means of identifying novel therapeutics to either prevent or combat these infectious diseases. The development of humanized mouse models offers a preclinical in vivo platform for further characterization of human viral infections and human immune responses triggered by these virus particles. Multiple strains of immunocompromised mice reconstituted with a human immune system and/or human hepatocytes are susceptible to infectious pathogens as evidenced by establishment of full viral life cycles in hope of investigating viral–host interactions observed in patients and discovering potential immunotherapies. This review highlights recent progress in utilizing humanized mice to decipher human specific immune responses against viral tropism.

## 1. Introduction

The use of small animal models such as mice and rats has contributed greatly to the understanding of disease pathogenesis and development of therapeutic approaches. Basically, these animals act as surrogates in representing human biology due to the limitations and ethical restrictions of obtaining tissue samples directly from human donors for research purpose. Moreover, these mammalian model systems are often easier to maintain and handle due to their nature of being small, have a high reproductive turnover, and share similar genomic and physiological characteristics with that of a human. Despite utilizing these amazing properties for basic biology, a fine line still separates mice studies from humans as they lack an integral component required in the human microenvironment herein, the immune system. For instance, it is known that the innate immune responses differ between man and mouse whereby mice lack a functional Toll-like-receptor 10 (TLR10) whereas TLR11, TLR12, and TLR13 which are expressed in mice are actually absent in the human genome [1,2]. Moreover, immune responses in wild-type mice infected with murine-adapted viruses are completely different to human immune responses triggered by human-specific pathogens due to inter-species diversity although they are used in studying the same virology [3,4,5]. Therefore, there is a huge demand for an improved small animal preclinical model to better recapitulate the human biological systems.

“Humanized” mice with functional human cell/tissue engraftment have garnered some interest lately and are gradually being recognized as an in vivo prerequisite in bridging the gap from bench-to-cage-to-bedside. However, it was not until the early 2000s when immunodeficient mice bearing mutations at the interleukin-2 receptor common gamma chain (*IL-2rγ^null^*) were used for efficient human cells and tissues engraftment [6,7]. This proved to be a major breakthrough as the absence of *IL-2rγ *led to severe impairments in multiple cytokine complexes involving IL-2, IL-4, IL-7, IL-9, IL-15, and IL-21 signalling and ultimately profound T cell defect [8]. When these *IL-2rγ^null^* mice were backcrossed with either the protein kinase DNA activated catalytic polypeptide mutation (*Prkdc^scid^*/*scid*) or with recombination activating gene (*Rag*) 1 or 2 (*Rag1^null^* or *Rag2^null^*) mutations, murine adaptive (T and B cells) and innate natural killer (NK) cells immunity were completely compromised including defects in mouse macrophages and some dendritic cell subsets [3]. Listed below are some of the more common approaches of engrafting human immune systems by using these immunodeficient *IL-2rγ^null^* recipients.

• Hu-PBL-SCID

This humanized mouse model is generated by injection of human peripheral blood leukocytes (PBL), resulting in rapid human T cells engraftment particularly CD3^+^ T cells by the end of the first week. However, the lifespan of these mice is short with majority of them succumbing to lethal xenogeneic graft-versus-host disease (GVHD). Nevertheless, such experimental window can be prolonged in NOD.Cg-*Prkdc^scid^Il-2rγ^tm1Wjl^* (NSG) strains deficient in murine MHC class I or II [3,9].

• Hu-SRC-SCID

A complete human immune system can be established in this mouse model through intrafemoral injection of CD34^+^ haematopoietic stem cells (HSCs) derived from bone marrow, cord blood, fetal liver or peripheral blood. These mice exhibit decent reconstitution of B cells, T cells, myeloid cells and antigen presenting cells (APCs) in the peripheral haematopoietic tissues albeit much lower levels of granulocytes, platelets and red blood cell development. Although T cells generated in the mouse thymus are not HLA-restricted, they may not recapitulate human T cell function entirely due to the lack of human-specific growth factors or cytokines [10,11].

• Bone marrow, liver and thymus (BLT) model

Bone marrow, liver and thymus (BLT) are used in generating this humanized mouse model. Both human fetal liver and thymus are transplanted under renal capsule whereas autologous fetal liver CD34^+^ HSCs were injected intravenously [12]. Like the Hu-SRC-SCID model, BLT mice develop all human lineages however they are HLA-restricted which will give rise to GVHD-like phenotype and ultimately restricts timeframe for animal experimentations (usually 25–30 weeks post-transplantation) [5,7].

Although the above humanized mouse models provide a solid foundation to elucidate human immune responses against viral infections, the study of hepatotropic pathogens remained elusive due to the lack of human hepatocytes in the liver. Therefore, development of human liver chimeric mouse models has allowed more efficient targeting of liver-specific viruses to mimic viral hepatitis in patients. Two commonly used humanized mouse liver models were established through destruction of mouse hepatocytes to permit repopulation of human ones. The first model utilizes albumin-urokinase plasminogen activator (Alb-uPA) transgenic mice whereby constitutive expression of uPA causes mouse hepatic injury to facilitate liver regeneration or expansion of human hepatocytes following transplantation [13]. The second model involves targeting of fumarylacetoacetate hydrolase (Fah) which is an enzyme required for mouse liver metabolism [14]. *Fah* knockout (KO) led to accumulation of toxic metabolites and hepatocellular injury to similarly allow expansion of transplanted human hepatocytes. In addition, *Fah* mutation was further crossed with *Rag2^null^*/*IL-2rγ^null^* mice as this mouse background has been reported to be an excellent recipient of human haematopoietic xenografts resulting in *Fah^null^*/*Rag2^null^*/*IL-2rγ^null^*(*FRG*) triple mutants [14,15]. One advantage of using these mice is that the extent of mouse liver damage can be controlled by administration of 2-(2-nitro-4- trifluoro-methylbenzoyl)-1,3-cyclohexanedione (NTBC) in mouse drinking water which replaces the loss of functional *Fah*. Thus, viral hepatitis can be easily characterized in livers of these humanized mouse models.

The popularity of using humanized mice instead of wild-type mice for in vivo validation coincides with marked improvements in supporting engraftments of human cells, tissues and the immune system. Therefore, humanized mice are slowly being utilized as preclinical gold standards for drugs evaluation and translational research. This not only led to a well-defined characterization of human-specific diseases such as acute infection and cancer but also better understanding of mechanisms underlying human biological responses against these diseases. In fact, many research groups worldwide have already showcased some of the advantages of humanized mice in studying human-specific infectious pathogens and immune responses.

## 2. Human Viral Infections in Humanized Mice

The human chimeric mouse model offers great opportunities to study human-specific infections such as viral replication and pathogenesis in the vicinity of the human immune system. The use of immunodeficient mice for human cells’ engraftment facilitates propagation of the human immune network allowing them to harbour species-specific viruses and exert species-specific responses. Therefore, it is important to take into consideration different mouse strains and engraftment methods in selecting the most optimal preclinical platform encompassing the virology of interest. Some of the more common infectious pathogens being studied in humanized mice are outlined below.

### 2.1. Human Immunodeficiency Virus (HIV)

Before the emergence of humanized mice, chimpanzees were the only animal model that permits infection of the retrovirus, HIV [16,17]. HIV, a single-stranded enveloped RNA virus, primarily infects human CD4^+^ T cells, macrophages and dendritic cells which ultimately causes acquired immunodeficiency syndrome (AIDS) over time if left untreated [18]. All 3 humanized mouse models mentioned earlier support HIV infection, but the BLT mice were frequently used due to a more complete human mucosal system for the study of vaginal and rectal transmission of HIV [18,19,20]. In addition, these humanized mice have allowed evaluation of many prophylactic drugs, anti-HIV antibodies, and cellular therapeutics for HIV replication prevention and elimination. Although it is known that HIV expresses essential proteins for virus production in culture, it was not until accessory proteins like viral infectivity factor (vif), viral protein R (vpr), viral protein U (vpu), and negative factor (nef) were identified in humanized mice as key players in modulating intrinsic immunity [21] that it became possible to investigate this.

The Hu-PBL-SCID mouse model was one of the earlier in vivo platforms used to understand the roles of HIV replication and pathogenesis [22,23]. For example, HIV-infected hu-PBL-SCID mice were used to assess the efficacy of HIV vaccinations in humans. PBLs that blocked HIV replication through antigen-specific human antibodies generated by these humanized chimeric mice following secondary immunization would demonstrate protective donor immunity. Since then, HSC-engrafted BLT mice coupled with infection of CCR5/CXCR4-tropic HIV strains have recapitulated human disease resulting in establishment of latent HIV reservoirs which exhibited neuropathology and central nervous system (CNS) infection [24,25,26,27]. Moreover, NOD-SCID BLT and NSG-BLT mice demonstrated retroviral spread in vivo supporting the notion of cell-to-cell trans-infection and that anti-retroviral therapy (ART) can indeed reduce the rate of HIV transmission [28,29]. However, some reports indicated that ART may not necessary clear the HIV reservoir entirely, as evidenced by presence of latently infected CD4^+^ T cells [24]. Other efforts involve knockdown of CCR5 in HSCs by using zinc finger nuclease and lentivirus technologies prior to transplantation into NSG mice [30,31]. These humanized mice portrayed resistance against CCR5-tropic HIV strain with reduced viral load and persistence in human CD4^+^ T cells function. Besides that, lentiviral transduction of HIV-specific CD8^+^ T cell receptors (TCRs) in HSCs following human engraftment has also displayed marked reduction in virus replication, increased viability of CD4^+^ T cells, and expansion of SL9-specific effector CD8^+^ T cells [32]. Similarly, treatment with anti-PD-1 antibody activated CD8^+^ T cells resulting in enhanced survivability of total T cells and decreased HIV plasma loads in BLT mice [33]. Overall, the recent increase in usage of humanized mice for HIV-related studies suggests that they are being accepted more readily by the research community as a preclinical alternative to other animal models in potentially identifying novel therapeutic interventions.

### 2.2. Dengue Virus (DENV)

Dengue virus (DENV) is a type of positive-stranded RNA virus (genus *Flavivirus*) transmitted by mosquitoes which can cause high fever, nausea, muscular and joint pains, etc. however, in some cases, the disease becomes deadly resulting in dengue haemorrhagic fever and dengue shock syndrome [34]. There are currently no dengue vaccines available due to the difficulty of incorporating all virus serotypes into a single preparation and the lack of suitable animal models that supports the infection. Although many had reported successful inoculation of DENV in wild-type mice, none were able to exhibit clinical symptoms observed in humans due to the natural resistance of mice against DENV infection through inhibition of interferon (IFN) signalling [35]. To overcome this, AG129 mice which lacked Type I and II IFN receptors were commonly used to harbor mouse-adapted DENV-2 viral replication and ultimately antiviral drugs screening. However, in vivo testing of DENV vaccines proved to be controversial as mouse-adapted DENV-2 strains are not naturally found in humans. Hence, humanized mice were utilized to overcome such issue and that DENV infection indeed triggered fever- and rash-like symptoms, thrombocytopenia, production of IgM anti-DENV antibodies and interferon-γ (IFN-γ)-producing T cell responses as seen in humans [36,37,38,39,40,41]. In fact, detectable levels of DENV were also present in spleen, bone marrow and liver of HSC-engrafted and BLT humanized mouse models. Further mechanistic insight on specific human lineages in humanized mice identified the importance of human NK cells as an early defence against DENV infection through secretion of IFN-γ [42].

### 2.3. Other Flaviviruses

Members of the virus family Flaviviridae other then DENV include yellow fever virus (YFV), west nile virus (WNV), Japanese encephalitis virus (JEV), tick-borne encephalitis virus (TBEV), and Zika virus [35]. Although non-human primates (NHPs) like Rhesus and/or cynomolgus macaques remain the best model for hosting DENV, YFV and Zika viruses naturally for vaccine development, the lack of resources in assessing immune responses such as detection of antigen-specific T cell responses after vaccination proved to be a limitation in NHP models. Therefore, humanized mice were utilized not only for antibody production but also immunophenotyping of specific immune subsets. In fact, humanized monoclonal antibodies have been reported for viral neutralization but not clearance against YFV [43,44] and JEV [45] in vivo. On the other hand, both WNV [46] and Zika virus [47,48,49] tackled the advantages of the human immune system in humanized BLT mice for further viral characterization and antiviral therapeutics. Notably, these BLT mice displayed persistence Zika viremia of up to 7 months post-infection which was attenuated with neutralizing antibody [49]. The study of TBEV remains to be investigated in humanized mice.

### 2.4. Epstein Barr Virus (EBV)

Epstein Barr virus (EBV) (also knowns as human herpesvirus 4; genus *Lymphocryptovirus*) is one of the most common viruses found to exclusively infect humans worldwide. EBV primarily infects B cells and epithelial cells which can be transmitted via saliva and genital secretions [50]. Infection is associated with various cancers (Hodgkin’s and Burkitt’s lymphomas, gastric and nasopharyngeal carcinomas, etc.) and autoimmune diseases (systemic lupus erythematosus, rheumatoid arthritis, multiple sclerosis, etc.). EBV infection can be described as a type of programmed virus that differentiates infected host cells into memory B cells for long-term persistence before reactivation into lytic replication via the mucosal surfaces prior to transmission of new hosts [51]. Multiple humanized mouse models were able to recapitulate EBV infection up to viral maintenance and transformation but not amplification when shedding into the saliva of the uninfected recipient [51,52,53,54]. This was demonstrated by humanized mice transplanted with CD34^+^ HSCs (in NSG, NOD-Rag1^null^IL2rg^null^ (NRG) and BALB/cA RAG2^null^IL2rg^null^ (BRG) background) as well as BLT mice in recapitulating the different EBV latency infection programs particularly in naïve B cells of healthy EBV carriers found in virus associated B cell lymphomas but do not exhibit final lytic replication in oropharyngeal epithelial cells. The proportion of EBV infected cells are generally controlled by CD8^+^ T cells but any remaining resistant population may potentially transform into tumors of which was observed in both Hu-PBL-SCID and Hu-SRC-SCID models [3,51]. Furthermore, EBV infected humanized mice developed HLA class I-dependent CD8^+^ T cell responses [55,56] and HLA-A2-restricted EBV epitope-specific responses [57]. Depletion of CD4^+^ and CD8^+^ T cells alone or in combination led to a surge in viral load and promotes tumorigenesis in vivo. However, immune surveillance in humanized mice was partially restored via blockade of inhibitory receptors PD-1 and CTLA4 suggesting possible vaccine formulations targeting cytotoxic lymphocytes [58].

In addition to T cell responses, NK cells also play an important role in EBV control as evidenced by increase in virus titer and augmented lytic infection following NK cells depletion [59]. In fact, immature NKG2A^+^KIR^-^NK cells expand in humanized mice 4 weeks post-EBV infection which was also detected in children with infectious mononucleosis [60]. Elevation of EBV infection ultimately led to formation of lymphoma which was highly dependent on the frequency of transformed B cells. Since EBV infection is human-specific and does not cross-react with murine species, consequences of lymphomagenesis observed in humanized mice are often associated with infectious viral particles residing majorly in spleen, mesenteric lymph node, kidney, liver and the peritoneal cavity which was also observed in patients [51].

### 2.5. Influenza

The flu virus that makes up part genera of the family *Orthomyxoviridae* is one of the most common causes of human respiratory infections [61]. Although vaccines and antiviral drugs are available to prevent or treat influenza respectively, there is no guarantee that one could escape infection entirely due to the constant evolution of different viral strains. Some of the mild symptoms include high fever, runny nose, coughing, sneezing, sore throats, etc. but complications may lead to more severe outcomes like gastroenteritis, pneumonia and even deaths. The use of small animal models like humanized mice would allow further understanding of influenza viral life cycle as well as viral replication which ultimately led to some of these symptoms. Indeed, humanized mice represent the best model for studying the flu virus but its poor development in the myeloid compartment remains a major drawback for triggering an immune response at mucosal surfaces such as the lungs. Many reports have demonstrated that human cytokines, interleukin-3 (IL-3) and granulocyte-macrophage colony-stimulating factor (GM-CSF) are essential for pulmonary homeostasis, myeloid cell development and host defence against pathogens [62,63,64]. Thus, immunodeficient mice transplanted with CD34^+^ HSCs were generated with human cytokines knock-in of IL-3 and GM-CSF to compete against mouse cytokines [62]. Substantial improvements in the development of alveolar macrophages triggered effective innate responses when challenged with influenza virus. These humanized mice consistently express high amounts of GM-CSF, tumor necrosis factor alpha (TNF-α), and IL-6 mRNA in the lungs after infection. In addition, M-CSF treatment following influenza infection in NSG Hu-SRC-SCID mice similarly displayed decreased viral transcripts which was associated with overproduction of proinflammatory cytokines, TNF-α and IL-6 [65]. One report also documented the importance of Flu-M1-specific CD8^+^ T cells and human dendritic cells in response to influenza-infected Hu-SRC NOD-*scidβ2m^null^* mice engrafted with autologous T cells from mobilized HSCs [66].

### 2.6. Ebola Virus

Ebola virus disease or EVD (caused by the genus *Ebolavirus*) is a type of viral haemorrhagic fever that can cause severe organ failure and ultimately death within an average of 2 weeks after detection of first symptoms due to low blood pressure and significant loss of bodily fluids. Although many groups could replicate similar haemorrhagic fever-like symptoms in wild-type mice infected with adapted Ebola virus, they failed to harbour Ebola virus when infected with EVD derived from human patients [67]. The introduction of humanized mice overcame this hurdle as engraftments of functional human T cells, B cells, NK cells, dendritic cells and macrophages triggered overactivation of the human immune and proinflammatory responses which facilitated host tissue injury [68]. It was recently reported that 6 days post EVD-infected BLT mice exhibited rapid virus dissemination and replication with high viral genome equivalents present in blood and tissue samples. In contrast, immunodeficient NSG mice were negative for Ebola virus even after 9 days of infection. Histopathological analyses of livers of EVD-infected BLT mice displayed severe liver inflammation and aggregates of necrotic hepatocytes which was dominated by neutrophils and macrophages. Production of viral nucleocapsids were also evident in liver tissues. Like patients with EVD, these mice portrayed significant elevation of proinflammatory mediators such as interferon alpha 2 (IFN-α2), TNF-α, IL-1α, IL-15, IL-1 receptor A (IL-1RA), monocyte chemoattractant protein 1 (MCP-1), IFN-γ–inducible protein 10 (IP-10) and granulocyte colony-stimulating factor (G-CSF). Similarly, NSG-HLA-A2 transgenic mice engrafted with HLA-A2 HSCs showed features of high viremia levels, liver steatosis and viral haemorrhagic fevers following Ebola virus inoculation [69].

### 2.7. Hantavirus

Hantavirus, a negative-sense RNA virus in the Hantaviridae family, is another type of infectious disease that can cause haemorrhagic fever with renal syndrome (HFRS) and hantavirus cardiopulmonary syndrome (HCPS) which give rise to increased vascular permeability and loss of platelets [70]. Rodents generally serve as natural hosts for hantaviruses but it can be detrimental in humans. Hantavirus replication was observed in cell culture but does not have cytopathic consequences suggesting that the human immune system is required for induction of HFRS or HCPS. Further reports confirmed a direct link between infection and human immune system as evidenced by damages on endothelial cell barrier functions which led to downstream respiratory disorders [71]. Humanized mice proved to be the most representative preclinical model in developing hantavirus-induced pathogenesis. HSC-engrafted NSG and NSG-HLA-A2 (engrafted with HLA-A2 HSCs) transgenic mice that were infected with hantavirus displayed high copy numbers of viral genomes in sera and other organs, particularly the lungs [70]. Pulmonary inflammation, progressive loss in body weight and drop in human platelet count quickly followed. In addition, the association of functional human CD8^+^ T cells and hantaviruses was attributed to the interaction with infected endothelial cells resulting in microvascular leakage along the alveolar walls. In contrast, hantavirus-infected immunodeficient mice were normal.

## 3. Hepatotropic Pathogens in Human Liver Chimeric Mouse Models

The immune system is essential for human immune surveillance for viral replication and providing specific immune biomarkers for viral clearance of all human pathogens. Listed above are some of the human-associated viruses known to have been investigated in humanized mice. However, the study of hepatotropic pathogens remained elusive due to the lack of human hepatocytes in the mouse liver. As mentioned earlier in this review, the establishment of two commonly used human liver chimeric mouse models, Alb-uPA transgenic mice and *FRG* KO mice, fully support the viral life cycle which led to further characterization of infectious pathogens [13,14,15]. In addition to humanization of mouse livers, some had also incorporated human immune system in order to study immune responses triggered by these hepatotropic infections. Examples of liver-associated pathogens utilizing both hepatocyte alone and hepatocyte with immune system dual humanization mouse model system are discussed below.

### 3.1. Hepatitis C Virus (HCV)

Hepatits C virus (HCV) is a single-stranded enveloped flavivirus that binds to cell surface in order to release virus particles into cells by receptor-mediated endocytosis [72,73]. HCV primarily affects the liver, but symptoms are typically mild or close to none. In most cases, the host immune system decides if HCV should be cleared completely or continue to persist. However, patients exposed to the virus would develop a chronic infection which can be defined as detectable viral replication for at least 6 months. Over time, chronic infection progressively induces hepatic cirrhosis and/or liver inflammation leading to the development of hepatocellular carcinoma (HCC). The restriction of HCV tropism to humans proved to be a major obstacle in understanding viral–host interactions, HCV-specific immune responses, disease progression, and identification of novel drug candidates. In the early 2000s, HCV replicons derived from the JFH-1 genotype 2a consensus sequence of a patient with fulminant hepatitis paved ways for in vitro production of HCV through robust RNA replication in cell lines including Huh7, HepG2, IMY-N9, and immortalized human hepatocytes as well as mouse cells such as MMHD3, MMH1-1, AML12, and NIH3T3 [74,75,76,77]. The identification of adaptive mutations at core, E1, E2, p7, NS2, NS5A coding regions further enhanced virus titers in both in vitro and in vivo (human liver chimeric mice) [78,79,80]. The Alb-uPA and *FRG* KO chimeric mice mentioned earlier were first models to exhibit localization of HCV viral proteins in human hepatocytes nodules [73,81,82]. HCV from mouse serum was also serially passaged and infected into 3 generations of mice confirming both synthesis and release of infectious viral particles [81]. Furthermore, Claudin-1 monoclonal antibody successfully inhibit HCV entry, cell–cell transmission and virus-induced signalling which led to clearance of persistent HCV infection in Alb-uPA mice [83]. In addition, HCV entry factors CD81 and human occludin (OCLN) were identified as key proteins required to render mice more susceptible to HCV infection [84,85,86]. Transgenic mice expressing these key human-specific factors facilitated viral uptake and replication which was suppressed by both innate and adaptive immunity in vivo.

To better understand the role of the human immune system in targeting HCV, humanized mice generated by co-transplantation of CD34^+^ HSCs and hepatic progenitors into hAlb-FKBP-Caspase 8 (*AFC8*)^+^ transgenic mice supported HCV-induced immune responses and liver diseases [87]. HCV-infected mice displayed elevated levels of human CD45^+^ leukocytes including CD68^+^ macrophages and CD3^+^ T cells infiltration in the liver. These mice also developed severe liver fibrosis but not in immunocompromised Alb-uPA and *FRG* KO chimeric mice after HCV infection indicating the importance of having a fully functional immune system to trigger liver damage [14,81,87]. Similarly, our group developed a new human immune and liver (HIL) mouse model by transplanting human fetal liver CD34^+^ cells which contain both hepatic progenitors and HSCs into NSG mice [88,89]. HCV-associated liver pathogenesis seen in patients was also confirmed in HIL mice. In fact, proinflammatory human cytokines such as B lymphocyte chemoattractant (BLC), IL-8, monocyte chemotactic protein-1 (MCP1), macrophage inflammatory protein-1 beta (MIP1b) and tumor necrosis factor receptor II (TNF-RII) were significantly upregulated in HIL mice just like in HCV-infected patients. Besides that, antibody-based depletion of specific human immune cells subsets such as CD4^+^, CD8^+^ T cells and CD14^+^ macrophages in HCV-infected humanized mice blocked progression of liver disease even though HCV RNA was still present. Furthermore, chronically infected HIL mice (up to 28 weeks) exhibited higher frequency of liver fibrosis, granulomatous inflammation and tumor formation in the form of HCC [90].

In addition to roles of human T cells in HCV-induced liver pathogenesis, NK cells were thought to possess antiviral immune response via the interaction of killer cell immunoglobulin like receptor, three Ig domains and short cytoplasmic tail 1 (KIR3DS1) and its ligand, HLA-F which in turn activates NK cells for resolution of HCV infection [91,92,93]. The recent discovery of KIR3DS1^+^ NK cells provides a novel insight in harnessing interaction between KIR3DS1 and HLA-F as a potential immunotherapeutic tool not only in HCV but other infectious diseases and cancer.

### 3.2. Hepatitis B Virus (HBV)

Like HCV infection, chronic HBV (a member of the Hepadnaviridae family; genus *Orthohepadnavirus*) can also cause liver inflammation/fibrosis which gives rise to liver cirrhosis and/or ultimately HCC in patients. Although HBV vaccines are available, it is not a solution for established infections [94,95]. The majority of antiviral therapies could suppress viral replication but not eradicate HBV entirely due to the stability of the covalently closed circular DNA (cccDNA) [96]. HBV cccDNA serves as a template for viral DNA replication in the cytoplasm of infected human hepatocytes through synthesis of four RNA transcripts, pregenomic RNA (pgRNA), preS, S, and X RNAs [97,98,99]. HBV DNA containing core particles would assemble with envelope proteins followed by secretion into the bloodstream. In fact, HBV-infected hepatocytes exhibited ~50 fold amplification of cccDNA at an early stage of infection in vitro which led to persistence production of virus particles even in the absence of stable integration of viral DNA [100]. Moreover, extensive studies have demonstrated that resolution of HBV cccDNA is primarily mediated by the killing of HBV-infected hepatocytes via cytotoxic T cells and noncytolytic mechanisms induced by cytokines [101,102,103]. HBV-specific CD8^+^ T cell-derived antiviral cytokines not only inhibited formation of pgRNA-containing capsids which subsequently blocked maturation into full length HBV DNA but also led to possible depletion of cccDNA [102].

Indeed, human liver chimeric mice remain the gold standard for supporting hepatotropic infections; however, the highly immunocompromised status of these engrafted mice precludes liver pathogenesis mediated by human immune surveillance. Hence, dual humanization of liver and the immune system was established in immunodeficient mice to study immune responses and liver disease progression in the context of HBV infection [104,105,106,107,108,109,110]. In one study, dual humanized mice were developed by injection of human fetal liver progenitor cells and HSCs from the same donor directly into the liver of NSG pups expressing human HLA-A2 (A2/NSG-hu) whereby the presence of HLA-A2 transgene enhances development of human MHC-restricted T lymphocytes [57,107]. Animals were then treated with murine-specific anti-Fas agonistic antibody (Jo2) to induce mouse liver damage for repopulation of human liver cells [107]. This dual humanized mouse model system allowed persistent HBV infection over several months followed by liver inflammation and fibrosis facilitated by M2-like macrophage infiltration suggesting a critical role for macrophage polarization in HBV-induced impairment and liver pathology. Similarly, another group demonstrated dual humanization of liver and immune system by syngeneic engraftment of human hepatoblasts and HSCs in *Fah* KO NOD *Rag1^null^*/* IL-2rγ^null^* (FNRG) mice which also portrayed rapid and sustained viremia upon HBV infection [108]. Although these humanized mice predominantly developed human T and B lymphocytes, administration of oncostatin-M further enhanced engraftment of human hepatoblasts and development of human monocytes and NK cells for the maintenance of HBV. Most recently, we proved that intrahepatic CD14^+^ myeloid cells contributed to chronic liver inflammation in both patients with viral-related end-stage disease and HBV infected HIL mice [111]. In addition, the severe liver inflammation mediated by proinflammatory CD14^+^HLA-DR^hi^CD206^+^ myeloid cells can be attenuated by treatment with oral antibiotics in HBV-infected HIL mice indicating that liver pathogenesis and intestine-derived bacterial products are linked.

Due to the limited access to fetal tissues, an alternative method of dual humanization of the liver as well as the immune system was established via transplantation of mature hepatocytes and HSCs from different donors [105,109]. Here, HBV-infected humanized mice exhibited partial immune control over viral life cycle as evidenced by presence of antigen-specific IgGs and liver-infiltrating Kupffer cells, NK cells (CD69^+^) and PD-1^+^ effector memory T cells which was in line with immunopathology observed in patients with chronic HBV [109]. Plasma from these infected mice also displayed elevated levels of inflammatory and immune-suppressive cytokines, C-X-C motif chemokine ligand 10 (CXCL10) and IL-10 which correlated with the intrahepatic CD4^+^ T cells subset. In addition, administration of nucleoside analogue entecavir reduced viral loads and decreased liver inflammation. On a similar note, further characterization of HBV-specific T cell receptors (TCRs)-reprogrammed nonlytic T cells identified apolipoprotein B mRNA editing enzyme, catalytic polypeptide 3 (APOBEC3) as key activation marker in suppressing viral replication without lysis [112]. The selective activation of lymphotoxin β receptor (LTBR) signalling and APOBEC3 upon antigen recognition without triggering inflammatory events could be a potential therapeutic strategy for chronic HBV patients.

### 3.3. Hepatitis D Virus (HDV)

Hepatitis D virus (HDV), a small spherical enveloped virusoid, is a subviral satellite which requires the presence of HBV for the production of infectious virus particles [113,114,115]. Although dormant on its own, chronic HDV infection offers the most severe form of viral hepatitis due to its combinatorial infection properties with HBV otherwise known as superinfection. HDV shares a very similar viral entry mechanism with HBV, frequently resulting in liver fibrosis, cirrhosis and eventually HCC. Similarly, human liver chimeric mice were utilized as models to support HBV/HDV coinfection and superinfection. Interestingly, HDV superinfection hindered HBV replication in vivo evidenced by significantly lower HBV viremia and intrahepatic cccDNA in HBV alone-infected humanized mice. Furthermore, treatments with HBV entry inhibitor, Myrcludex-B (MyrB) and lonafarnib (LNF) alone or in combination suppressed virus titer efficiently but failed to cure HDV infection [114]. As the majority of liver pathogenesis in HDV patients is attributed to intrahepatic immunopathology, human liver chimeric mice on C57BL/6 background stably expressing HBV transgene were assessed against HDV infection. The murine immune system triggered activation of innate immunity through increased population subsets of NK cells, NK T cells, and mucosal-associated invariant T cells (MAIT) as well as elevated cytokines production of IL-23, IFN-γ, TNFα, IL-1b, IL-33, IFN-β, and GM-CSF. Human interferon-stimulated genes (ISGs) like (2’,5’-oligoadenylate synthetase-like proteins 1 and 2 (OAS1 & OAS2), IP-10, Interferon-induced GTP-binding protein Mx1, RNA-activated protein kinase R (PKR), interferon stimulated gene 15 (ISG15), and signal transducers and activators of transcription gene 1 (STAT1) were also highly expressed [114,116]. However, studies on human immune responses in HDV-superinfected humanized mice remains to be investigated.

### 3.4. Hepatitis E Virus (HEV)

Hepatitis E virus (HEV) is a single-stranded, non-enveloped RNA icosahedral virus comprising a positive-sense, single stranded RNA genome which transmits viruses via the faecal–oral route. Although acute HEV infection is quite common, patients often undergo full recovery following antiviral treatments. Animal studies of HEV infection are limited but human liver chimeric mice remain the best model for further viral characterization. Since only the mouse liver is humanized, administration of HEV had to be performed intravenously or via the mouse spleen rather than orally due to species-specific intestinal host restrictions [117,118,119,120]. A full viral life cycle was established in HEV-infected mice as evidenced by detection of viral RNA in faeces, bile and liver. Mice with humanized liver harboured the virus over several months without hepatotoxicity but displayed increased expression of innate genes like CXCL9, CXCL10, HLA class 1, and ISGs which may be mediated by the immune response [119]. However, just like HDV research, human-specific immune responses targeting HEV infection are yet to be explored in vivo. Nevertheless, antiviral therapeutics using Ribavirin have shown efficient reduction of virus titer in both plasma and faeces of HEV-infected human liver chimeric mice [118,119].

## 4. Limitations and Future Directions

The evolution of humanized mouse models has contributed to better characterization of human infectious diseases and immune responses. Only in recent times have humanized mice been utilized more frequently as a preclinical platform for in vivo validation, but key improvements in specific areas are required. Firstly, the study of murine-adapted viruses in wild-type mice may not necessarily recapitulate an infectious phenotype observed in clinical settings, suggesting that the source of species-specific virus inoculum is critical for fully understanding viral life cycles and replication/production capabilities in humanized mice. Therefore, the establishment of a fully functional human immune system and/or liver system in vivo is required to accommodate such human-specific infectious pathogens. Reconstitution of human immune cells permits the study of blood-borne viruses while repopulation of human hepatocytes in human liver chimeric mice is essential for the study of liver-associated infectious diseases like viral hepatitis. Most importantly, dual humanization of liver and immune system was able to recapitulate liver pathogenesis and trigger immune responses similarly observed in patients. Secondly, although human immune profiling can be investigated in humanized mice when challenged with infectious pathogens, the less optimal cell reconstitution and responses of the human innate immune components in humanized mice remain a major limiting factor [1,2]. New strains of immunocompromised mice have been generated by further replacement of murine immune counterpart components with human histocompatibility markers and other human specific molecules e.g. human cytokines via transgenic knock-in approaches to improve engraftment and maturation of human immune subsets [9,57,87,117]. Other efforts to improve human immune reconstitution in mice have been enforced through various methods like expansion and differentiation of CD34^+^ HSCs in vitro prior to mice injection [121], gene editing technology [122], packaging and delivery of human cytokines [123,124,125], etc. Hence, technological advancements in humanized mouse models have allowed more robust in vivo characterization to further elucidate viral–host interactions and identify novel immunotherapies and/or vaccine strategies.

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
