# Peer review of "Humanized Mouse Models for the Study of Infection and Pathogenesis of Human Viruses"

_viruses, 2018, doi:10.3390/v10110643_

Round 1

Reviewer 1 Report

Overall, this review is well structured, pertinent and interesting to read. There is a need for a review on humanised mice and the manuscript is very relevant. I do have a few minor suggestions/comments. 

It would be good to have all viruses introduced with a similar level of detail. Family and genus should be named and perhaps even the type of virus (RNA/DNA; positive/negative strand, etc.) It is introduced for some but not others.

The various paragraphs of the manuscript are sometimes written vaguely and additional details or information would be beneficial to the reader. I would suggested revising the manuscript and writing each section a little more in depth.

For example lines 76-77 say that GVDH-like phenotype ultimately restricts timeframe for animal experiments. How long is this time frame? 

Lines 125-126 authors describe that Hu-PBL-SCID were one of the earliest platforms used to understand HIV replication pathogenesis. Perhaps give more details about what was done or exactly what information was gathered by using these mice. 

Lines 150-152 explain that many reported successful inoculation of dengue in wild-type mice, none were able to exhibit clinical symptoms observed in humans. Perhaps describe what the wt mice showed or why they were used? However, to my knowledge, the gold standard is the AG129 mouse model which is not discussed. The issue with studying dengue in mice is that you need to use an immunocompromised mouse and this is not discussed in the paragraph. They just state a "lack of suitable animal models" but this could be expanded. 

Lines 177-180 say that multiple humanised mouse models were able to recapitulate EBV infection up to viral maintenance and transformation but not  amplification when shedding... But which mice? and how? 

On lines 39-40 it is stated that "human immune responses triggered by human-specific pathogens are completely different when compared to infectious agents of murine origin." This statement is not surprising. Do the authors mean for a same virus coming from human and mice or just pathogens in general?  Written as is, I am not sure that this is the correct argument for the next sentence.

 The authors write on line 162-163 that mice are not able to naturally host the infection. Whilst this is correct, it is also the case for dengue in the above section but not mentioned. Therefore, I am unsure that it is the adequate argument as to why there are limited studies using humanised mice platform for the other flaviviruses. 

In lines 202-203 it is stated that small animal models with immune cells are required to support infection and exert human immune responses. The authors are linking that statement with the previous sentences describing the symptoms and writing that treatment and vaccines are not always effective. I don't think that is the right argument for using humanised mice. The link seems incorrect. 

In lines 222-223, it says that many groups could replicate similar haemorrhagic fever-like symptoms in wt mice. However was that using wt virus or mouse adapted? 

I think the authors need to address the usefulness of the  JFH-1 clone in studying HCV. This was  important in earlier studies both in vitro and in vivo and was for a long time the only way to gather information about HCV pathogenesis.

When describing HBV, the authors state that "extensive studies have demonstrated that resolution of HBV cccDNA is primarily mediated by killing of HBV-infected hepatocytes via cytotoxic T-cells and noncytolytic mechanisms induced by cytokines." More details about the latter could be given even if just stating the main soluble factors involved. 

Finally, in section 4, limitations and future direction, it is unclear what the authors meant by "the  source of virus inoculum may affect infections and replication in humanised mice due to the majority of viruses being restricted to human tropism". do they mean source as in a clinical isolate, grown up in cell culture, an animal virus? Perhaps this could be clarified.

Author Response

We thank reviewer's comments and please find our responses attached

We thanked Reviewer 1 for his/her constructive comments. We would like to address all of Reviewer 1’s comments in italics below:
It would be good to have all viruses introduced with a similar level of detail. Family and genus should be named and perhaps even the type of virus (RNA/DNA; positive/negative strand, etc.) It is introduced for some but not others.

Response:
“a single-stranded enveloped RNA virus” was added in line 119.
“positive-stranded RNA virus (genus flavivirus)” was added in line 152.
“Members of the virus family Flaviviridae” was added in line 171.
“(also knowns as human herpesvirus 4; genus Lymphocryptovirus)” was added in line 185.
“virus that makes up part genera of the family Orthomyxoviridae” was added in line 222.
“(caused by the genus Ebolavirus)” was added in line 246.
“a negative-sense RNA virus in the Hantaviridae family” was added in line 271.
“(a member of the Hepadnaviridae family; genus Orthohepadnavirus)” was added in lines 349-350.
“a small spherical enveloped virusoid” was added in line 404.
“single-stranded” was added in line 425.

For example lines 76-77 say that GVDH-like phenotype ultimately restricts timeframe for animal experiments. How long is this time frame?

Response: “usually 25-30 weeks post-transplantation” was added in line 82.

Lines 125-126 authors describe that Hu-PBL-SCID were one of the earliest platforms used to understand HIV replication pathogenesis. Perhaps give more details about what was done or exactly what information was gathered by using these mice.

Response: “For example, HIV-infected hu-PBL-SCID mice were used to assess the efficacy of HIV vaccinations in humans. PBLs that blocked HIV replication through antigen-specific human antibodies generated by these humanized chimeric mice following secondary immunization would demonstrate protective donor immunity” was added in lines 130-133. References Moisier, D.E. et.al. and Tary-Lehmann, M. were added in support of this point.

Lines 150-152 explain that many reported successful inoculation of dengue in wild-type mice, none were able to exhibit clinical symptoms observed in humans. Perhaps describe what the wt mice showed or why they were used? However, to my knowledge, the gold standard is the AG129 mouse model which is not discussed. The issue with studying dengue in mice is that you need to use an immunocompromised mouse and this is not discussed in the paragraph. They just state a "lack of suitable animal models" but this could be expanded.

Response: “due to the natural resistance of mice against DENV infection through inhibition of interferon (IFN) signaling [35]. To overcome this, AG129 mice which lacked Type I and II IFN receptors were commonly used to harbor mouse-adapted DENV-2 viral replication and ultimately antiviral drugs screening. However, in vivo testing of DENV vaccines proved to be controversial as mouse-adapted DENV-2 strains are not naturally found in humans” was added in lines 158-162.

Lines 177-180 say that multiple humanised mouse models were able to recapitulate EBV infection up to viral maintenance and transformation but not amplification when shedding... But which mice? and how?

Response: “This was demonstrated by humanized mice transplanted with CD34+ HSCs (in NSG, NRG and BRG background) as well as BLT mice in recapitulating the different EBV latency infection programs particularly in naïve B cells of healthy EBV carriers found in virus associated B cell lymphomas but do not exhibit final lytic replication in oropharyngeal epithelial cells” was added in lines 200-204.

On lines 39-40 it is stated that "human immune responses triggered by human-specific pathogens are completely different when compared to infectious agents of murine origin." This statement is not
surprising. Do the authors mean for a same virus coming from human and mice or just pathogens in general? Written as is, I am not sure that this is the correct argument for the next sentence.

Response: “In addition, the human immune responses triggered by human-specific pathogens are completely different when compared to infectious agents of murine origin” was rephrased to “Moreover, immune responses in wild-type mice infected with murine-adapted viruses are incomparable to human immune responses triggered by human-specific pathogens due to inter-species differences although there are used in studying the same virology” in lines 39-41.

The authors write on line 162-163 that mice are not able to naturally host the infection. Whilst this is correct, it is also the case for dengue in the above section but not mentioned. Therefore, I am unsure that it is the adequate argument as to why there are limited studies using humanised mice platform for the other flaviviruses.

Response: “Only limited studies of such viruses were reported using the humanized mice platform possibly due to the fact that mice are not able to naturally host the infection” was rephrased to “Although non-human primates (NHPs) like Rhesus and/or cynomolgus macaques remain the best model for hosting DENV, YFV and Zika viruses naturally for vaccine development, the lack of resources in assessing immune responses such as detection of antigen-specific T cell responses after vaccination proved to be a limitation in NHP models. Therefore, humanized mice were utilized not only for antibody production but also immunophenotyping of specific immune subsets” in lines 173-177.

In lines 202-203 it is stated that small animal models with immune cells are required to support infection and exert human immune responses. The authors are linking that statement with the previous sentences describing the symptoms and writing that treatment and vaccines are not always effective. I don't think that is the right argument for using humanised mice. The link seems incorrect.

Response: “Therefore, small animal models with human immune cells are required to not only support infection of human pathogens but also exert effective human immune responses” was rephrased to “The usage of small animal models like humanized mice would allow further understanding of influenza viral life cycle as well as viral replication which ultimately led to some of these symptoms” in lines 227-229.

In lines 222-223, it says that many groups could replicate similar haemorrhagic fever-like symptoms in wt mice. However was that using wt virus or mouse adapted?
Response: “infected with adapted Ebola virus” was added in line 246.

I think the authors need to address the usefulness of the JFH-1 clone in studying HCV. This was important in earlier studies both in vitro and in vivo and was for a long time the only way to gather information about HCV pathogenesis.

Response: “In the early 2000s, HCV replicons derived from the JFH-1 genotype 2a consensus sequence of a patient with fulminant hepatitis paved ways for in vitro production of HCV through robust RNA replication in cell lines including Huh7, HepG2, IMY-N9, and immortalized human hepatocytes as well as mouse cells such as MMHD3, MMH1-1, AML12, and NIH3T3 [74-77]. The identification of adaptive mutations at core, E1, E2, p7, NS2, NS5A coding regions further enhanced virus titers in both in vitro and in vivo (human liver chimeric mice) [78-80]” was added in lines 308-314. References Kata, T., et.al., Date, T., et.al., Kanda, T., et.al., Uprichard, S.L., et.al., Zhong, J., et.al., Kaul, A., et.al., and Bukh, J. were added in support of this point.

When describing HBV, the authors state that "extensive studies have demonstrated that resolution of HBV cccDNA is primarily mediated by killing of HBV-infected hepatocytes via cytotoxic T-cells and noncytolytic mechanisms induced by cytokines." More details about the latter could be given even if just stating the main soluble factors involved.

Response: “HBV-specific CD8+ T cell-derived antiviral cytokines not only inhibited formation of pgRNA-containing capsids which subsequently blocked maturation into full length HBV DNA but also led to possible depletion of cccDNA [102]” was added to lines 361-364.

Finally, in section 4, limitations and future direction, it is unclear what the authors meant by "the source of virus inoculum may affect infections and replication in humanised mice due to the majority of viruses
being restricted to human tropism". do they mean source as in a clinical isolate, grown up in cell culture, an animal virus? Perhaps this could be clarified.

Response: “Firstly, source of virus inoculum may affect efficiency of infections and viral replication/production in humanized mice due to majority of viruses being restricted to human tropism” was rephrased to “Firstly, study of murine-adapted viruses in wildtype mice may not necessarily recapitulate infectious phenotype observed in clinical settings suggesting that the source of species-specific virus inoculum is critical for fully understanding viral life cycles and replication/production capabilities in humanized mice” in lines 443-446.

Reviewer 2 Report

This manuscript by Lai and Chen is well-written and has covered most major infectious diseases.

The only concern is that transgenic or knock-in mice expressing human genes are also considered "humanized mice" (Curr Opin Endocrinol Diabetes Obes. 2010 Apr; 17(2): 120–125.). The authors should at least discuss these.

Author Response

We thank reviewer's comments and please find our responses attached.

We thanked Reviewer 2 for his/her comment. We would like to address Reviewer 2’s comment in italics below:
This manuscript by Lai and Chen is well-written and has covered most major infectious diseases.
The only concern is that transgenic or knock-in mice expressing human genes are also considered "humanized mice" (Curr Opin Endocrinol Diabetes Obes. 2010 Apr; 17(2): 120–125.). The authors should at least discuss these.

Response: [We use a simple definition of humanized mice as “mice engrafted with functional human cells or tissues or expressing human transgenes.” Depending on the experimental question, different models of immunodeficient humanized mice are utilized.”] This was quoted from the review article of the attached journal. The definition of “humanized mice” varies in different context. In our manuscript, we defined “humanized mice” as mice reconstituted with functional human immune system and/or human liver chimeras in the study of human viral infections. We have add the clear definition in the introduction. We also outlined recent improvements in utilizing certain mouse strains (e.g. Alb-uPA, HLA-A2, and AFC8+ transgenic mice) to enhance engraftment of human cells/tissue to study activation of specific human immune subsets upon virus inoculation.
